# Model-based evaluation of school- and non-school-related measures to control the COVID-19 pandemic

Ganna Rozhnova [1,2 ✉], Christiaan H. van Dorp[3], Patricia Bruijning-Verhagen [1], Martin C. J. Bootsma [1,4], Janneke H. H. M. van de Wijgert [1,5], Marc J. M. Bonten[1,6] & Mirjam E. Kretzschmar [1]

The role of school-based contacts in the epidemiology of SARS-CoV-2 is incompletely understood. We use an age-structured transmission model fitted to age-specific seroprevalence and hospital admission data to assess the effects of school-based measures at different time points during the COVID-19 pandemic in the Netherlands. Our analyses suggest that the impact of measures reducing school-based contacts depends on the remaining opportunities to reduce non-school-based contacts. If opportunities to reduce the effective reproduction number ($R_e$) with non-school-based measures are exhausted or undesired and $R_e$ is still close to 1, the additional benefit of school-based measures may be considerable, particularly among older school children. As two examples, we demonstrate that keeping schools closed after the summer holidays in 2020, in the absence of other measures, would not have prevented the second pandemic wave in autumn 2020 but closing schools in November 2020 could have reduced $R_e$ below 1, with unchanged non-school-based contacts.

[1] Julius Center for Health Sciences and Primary Care, University Medical Center Utrecht, Utrecht University, Utrecht, The Netherlands. [2] BioISI—Biosystems & Integrative Sciences Institute, Faculdade de Ciências, Universidade de Lisboa, Lisboa, Portugal. [3] Theoretical Biology and Biophysics (T-6), Los Alamos National Laboratory, Los Alamos, NM, USA. [4] Mathematical Institute, Faculty of Science, Utrecht University, Utrecht, The Netherlands. [5] The Institute of Infection, Veterinary and Ecological Sciences, University of Liverpool, Liverpool, UK. [6] Department of Medical Microbiology, University Medical Center Utrecht, Utrecht University, The Netherlands. ✉email: g.rozhnova@umcutrecht.nl

In autumn 2020, many countries, including the Netherlands, are experiencing a second wave of the COVID-19 pandemic[1]. During the first wave in spring 2020, general population-based control measures were introduced in the Netherlands, which involved physical distancing (including refraining from hand-shaking), frequent hand-washing and other hygiene measures, and self-quarantine when symptomatic. In addition, many public places and schools were closed. These contact-reduction measures were relaxed starting from May, and the incidence of COVID-19 started to increase again at the end of July[1]. From the end of August onwards, contact-reduction measures were intensified in a step-wise manner. Schools closed during July and August for summer break, reopened at the end of August, and have remained open until 16 December, with the exception of a one-week autumn break. Some measures were implemented in schools after the summer break to reduce transmission. Students and teachers in secondary schools have to wear masks when not seated at their desks, and students have to keep distance from teachers. A student with cold- or flu-like symptoms has to stay at home.

The step-wise increase in control measures after the summer started with earlier closing times of bars and restaurants, reinforcement of working at home (in September), followed by closure of all bars and restaurants, theaters, cinemas, and other cultural meeting places in November and obligatory mask wearing in all public places since 1 December. According to the National Institute for Public Health and the Environment (RIVM), estimated effective reproduction numbers ($R_e$) for the Netherlands were about 1.3 in the period 27 August–6 September and about 1.0 in the period 7–13 November[1]. The aim of measures implemented by the government in autumn 2020 was to reduce $R_e$ to 0.8. The failure to achieve this might be due to reduced societal acceptance of control measures, and/or due to the lack of school closure. The role of children and their contacts during school hours in the spread of SARS-CoV-2 is in fact not well understood[2,3]. In this study, we explored this role with a mathematical model fitted to COVID-19 data from the Netherlands.

Closure of schools is considered an effective strategy to contain an influenza pandemic[4], based on both model calculations and observational studies of the influence of school holidays on the spread of influenza[5,6]. The reasons for this are the high contact rates in young age groups[7] and the susceptibility of children and young people to the influenza virus. In contrast to influenza, children seem to be less susceptible to SARS-CoV-2 than adults and, based on sparse data, the susceptibility to SARS-CoV-2 increases with age[8,9].

In the absence of empirical SARS-CoV-2 data, mathematical modeling can help to quantify the role of different age groups in the distribution of SARS-CoV-2 in the population[10,11], and to evaluate the impact of interventions on transmission[12–17]. Such models can help to estimate the reduction in the effective reproduction number for different contact-reduction scenarios within or outside school environments. Model predictions about the relative epidemic impacts of school- and non-school-based measures can assist policymakers in selecting combinations of measures during different stages of the pandemic that optimally balance potential harms and benefits. Predictions generated by models that include differences in susceptibility and contact rates in different age groups can also aid in deciding which school age groups should be the primary target of school-based interventions.

We used an age-structured transmission model fitted in a Bayesian framework to age-specific hospital admission data (27 February–30 April 2020) and cross-sectional age-specific seroprevalence data (April/May 2020)[18] to evaluate the effects of control measures aimed at reducing school and other (non-school-related) contacts in society in general at different time points during the COVID-19 pandemic in the Netherlands. The model makes use of age-specific contacts rates before and after the first lockdown[19] and contact rates in schools[7,20], and accounts for different susceptibility to SARS-CoV-2 among younger, middle-aged, and older persons. Using the model equipped with parameter estimates, we provide a comparative study of the impact of school- and non-school-related measures on the effective reproduction number in August 2020, before the most recent set of measures was implemented, and in November 2020, when the most recent measures were still in place. In particular, we assess whether keeping schools closed after the summer holidays in 2020 would have prevented the second pandemic wave in the autumn and whether closing schools in November 2020 could have helped to achieve the control of the pandemic. We quantify reductions in $R_e$ due to closing schools for different ages and make recommendations on which school ages should be targeted to design effective school-based interventions.

## Results

**Epidemic dynamics**. The model shows a very good agreement between the estimated age-specific hospitalizations and the data (Fig. 1). The number of hospitalizations increases with age, with the highest peaks in hospitalizations observed in persons above 60 years old. The estimated probability of hospitalization increases nearly exponentially with age (as shown by an approximately linear relationship on the logarithmic scale, Fig. 2), except for persons under 30 years old, in whom the number of hospitalizations was low. The estimated probability of hospitalization increased from 0.09% (95% CrI 0.05–0.15%) in persons under 20 years old to 4.37% (95% CrI 2.80–8.82%) in persons older than 80 years (Supplementary Fig. 2).

The model accurately reproduces the percentage of seropositive persons distributed across the age groups (Fig. 3). The median seroprevalence in the population was 2.7%, with the maximum seroprevalence observed in persons between 20 and 40 years old (about 3.5%). The lowest seroprevalence was among children in the 0–10 years age group (0.9%). Note that if our model did not include age dependence of susceptibility to SARS-CoV-2, the seroprevalence peak would be expected among children because they have the largest numbers of contacts in the population.

The estimated probability of transmission per contact was 0.07 (95% CrI 0.05–0.12) before the first lockdown and it decreased by 48.84% (95% CrI 23.81–87.44%) after the first lockdown. The reduction in susceptibility relative to susceptibility in persons above 60 years old was 23% (95% CrI 20–28%) in persons under 20 years old and 61% (95% CrI 50–72%) for persons between 20 and 60 years old (Supplementary Fig. 3). The estimated basic reproduction number was 2.71 (95% CrI 2.15–5.18) in the absence of control measures (February 2020) (Supplementary Fig. 4a), and dropped to 0.62 (95% CrI 0.29–0.74) after the full lockdown (April 2020) (Supplementary Fig. 4b). Supplementary Figures 1–4 show an overview of all parameter estimates.

The joint posterior density of the estimated parameters reveals strong positive and negative correlations between some of the parameters (Supplementary Fig. 5). For instance, the initial fraction of infected individuals is negatively correlated with the probability of transmission per contact and the hospitalization rate, as a small initial density can be compensated by a faster growth rate or a larger hospitalization rate. For that reason, the age-specific hospitalization rates are all positively correlated. These correlations highlight the necessity of complementing the hospitalization time series data with seroprevalence data, even if the sample size of the latter is small. Without the seroprevalence data many parameters would be difficult to identify.

**School and non-school-based measures**. The sequence of measures implemented and lifted during the pandemic in the

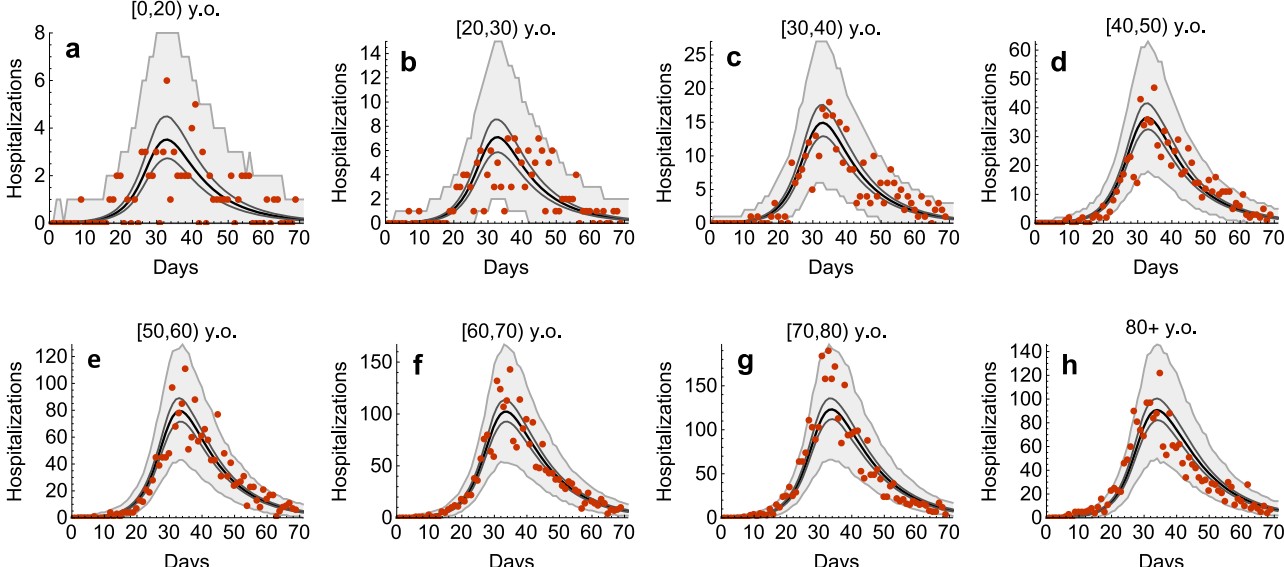

**Fig. 1 Estimated age-specific hospital admissions.** The black lines represent the estimated medians. The dark gray lines correspond to 95% credible intervals obtained from 2000 parameter samples from the posterior distribution, and the shaded regions show 95% Bayesian prediction intervals. The dots are daily hospitalization admission data (all data points are included). Day 1 corresponds to 22 February 2020 which is 5 days prior to the first officially notified case in the Netherlands (27 February 2020). Panels **a**–**h** refer to different age groups.

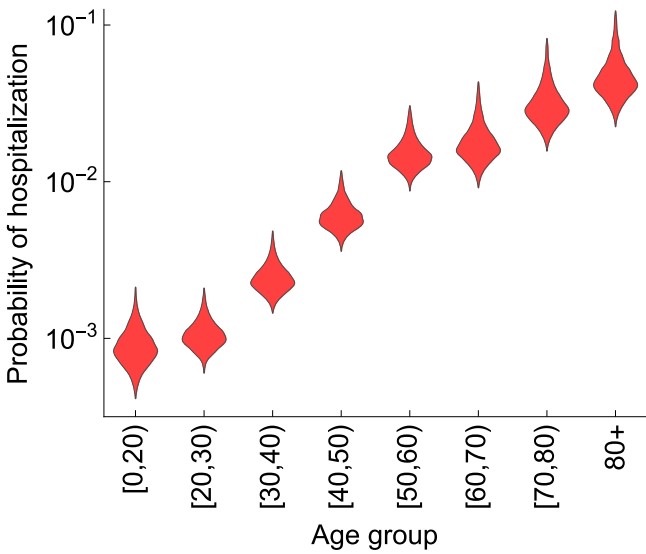

**Fig. 2 Estimated age-specific probability of hospitalization.** The violin shapes represent the marginal posterior distribution for 2000 samples of the probability of hospitalization in the model.

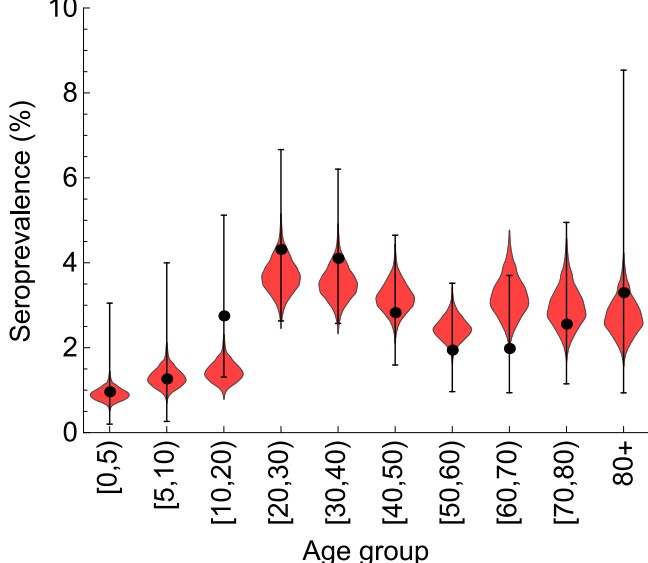

**Fig. 3 Estimated age-specific seroprevalence.** The data (dots) are shown as the percentage of seropositive persons based on a seroprevalence survey that was conducted in April/May 2020. The number of positive and total samples defining this percentage for each age category is supplied in seroprevalence data file accompanying this study (see "Data availability"). The error bars represent the 95% confidence (Jeffreys) interval of the percentage. The violin shapes represent the marginal posterior distribution for 2000 samples of the percentage of seropositive persons in the model.

Netherlands and the respective estimated values of the effective reproduction numbers are shown schematically in Fig. 4. We used the fitted model to separately determine the effect on the effective reproduction number of decreasing contacts in schools and of decreasing other (non-school-related) contacts in society in general in August 2020 (Fig. 5) and in November 2020 (Fig. 6). In doing so, we varied one type of contact and kept the other type constant. For each scenario, the reduction in contact rate was varied between 0 and 100%. The aim of reducing the number of contacts of each type is to decrease the effective reproduction number below 1.

We first considered the situation in August 2020 (Fig. 5), when schools had just opened after the summer holidays and when control measures in the population were less stringent than in

April (full lockdown). Between August and December 2020, the only infection prevention measure in primary schools was the advice to teachers and pupils to stay at home in case of symptoms or a household member diagnosed with SARS-CoV-2 infection; physical distancing between teachers and pupils (but not between pupils) only applied to secondary schools. We therefore assumed that the effective number of contacts in schools was the same as before the pandemic. For other (non-school-related) contacts in

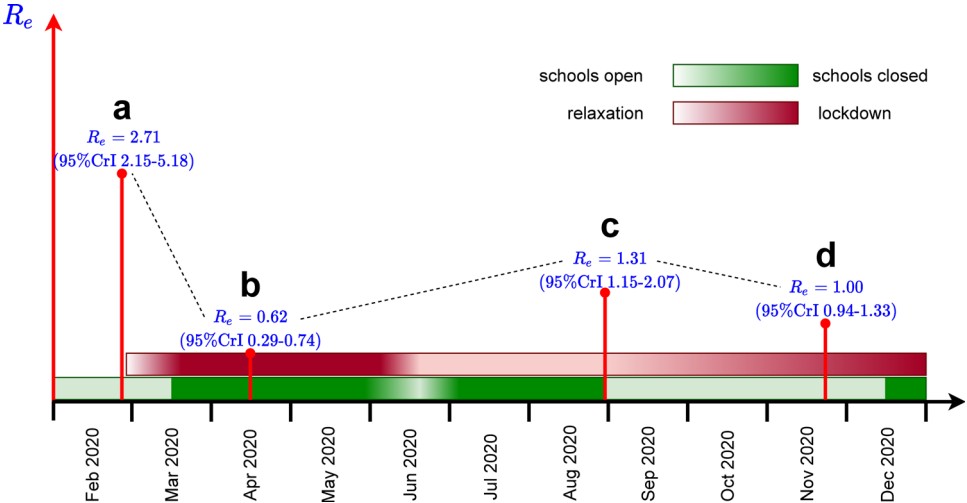

**Fig. 4 Schematic timeline of the pandemic in the Netherlands during 2020.** Outlined are times of the introduction and relaxation of control measures, and the estimated effective reproduction numbers for (**a**) start of the pandemic (February 2020), (**b**) full lockdown (April 2020), (**c**) schools opening (August 2020), and (**d**) partial lockdown (November 2020). See Supplementary Fig. 4 for the distributions of the reproduction numbers.

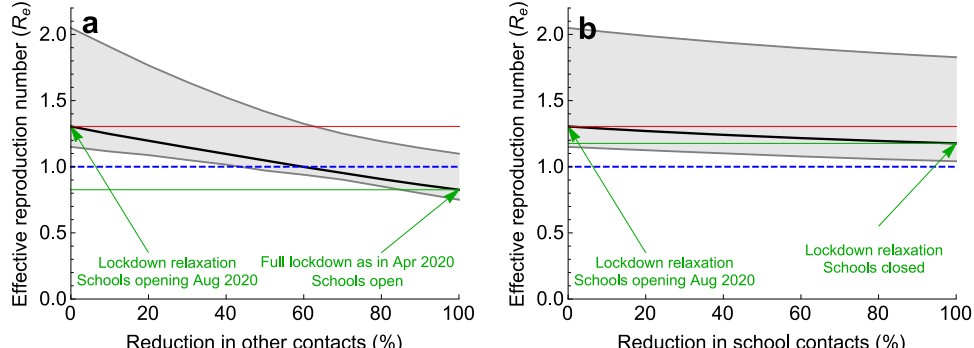

**Fig. 5 Impact of reduction of two types of contacts on the effective reproduction number in August 2020.** Percentage reduction in (**a**) other (non-school-related) contacts in society in general and (**b**) school contacts, with the number of the other type of contact kept constant in each of the two panels. The scenario with 0% reduction describes the situation in August 2020, when schools just opened in the Netherlands. The scenario with 100% reduction represents a scenario with either (**a**) maximum reduction in other (non-school-related) contacts in society in general to the level of April 2020 or (**b**) complete closure of schools. The solid black line describes the median and the shaded region represents the 95% credible intervals obtained from 2000 parameter samples from the posterior distribution. The red line is the starting value of $R_e$ (situation August 2020) and the green line is the value of $R_e$ achieved for 100% reduction in contacts. The blue line indicates $R_e$ of 1. To control the pandemic, $R_e < 1$ is necessary.

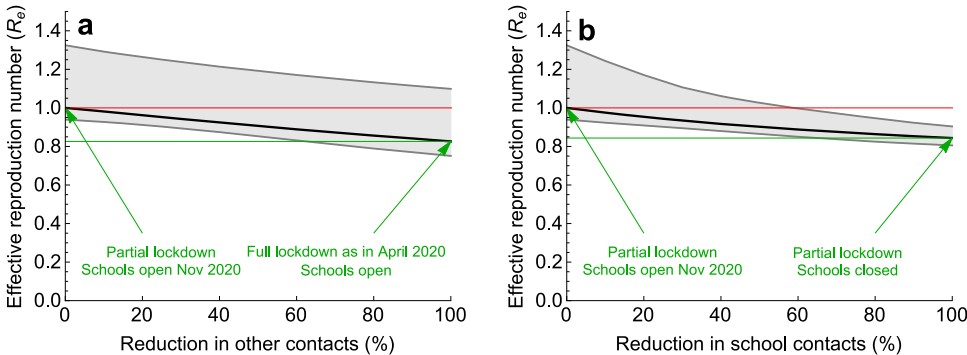

**Fig. 6 Impact of reduction of two types of contacts on the effective reproduction number in November 2020.** Percentage reduction in (**a**) other (non-school-related) contacts in society in general and (**b**) school contacts, with the number of the other type of contact kept constant in each of the two panels. The scenario with 0% reduction describes the situation in November 2020. The scenario with 100% reduction represents a scenario with either (**a**) maximum reduction in other (non-school-related) contacts in society in general to the level of April 2020 or (**b**) complete closure of schools. The solid black line describes the median and the shaded region represents the 95% credible intervals obtained from 2000 parameter samples from the posterior distribution. The red line is the starting value of $R_e$ (situation November 2020) and the green line is the value of $R_e$ achieved for 100% reduction in contacts. To control the pandemic, $R_e < 1$ is necessary.

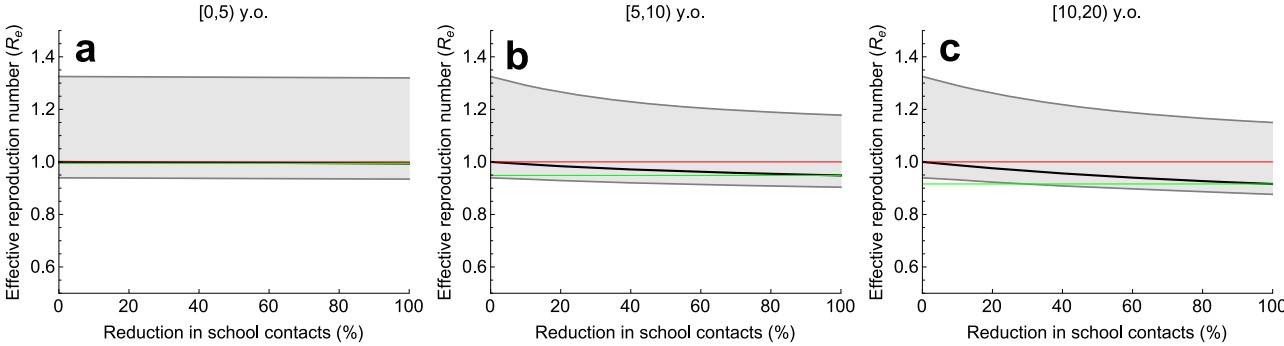

**Fig. 7 Impact of reduction of school contacts in different age groups on the effective reproduction number in November 2020.** Percentage reduction in school contacts among (**a**) [0, 5) years old, (**b**) [5, 10) years old, and (**c**) [10, 20) years old. In each panel, we varied the number of school contacts in one age group while keeping the number of school contacts in the other two age groups constant. The scenario with 0% reduction describes the situation in November 2020 with $R_e$ of about 1 (partial lockdown intended to prevent the second wave), where all schools are open without substantial additional measures. The reduction of 100% in school contacts represents a scenario with the structure of non-school contacts in society in general as in November 2020 and schools for students in a given age group closed. The solid black line describes the median and the shaded region represents the 95% credible intervals obtained from 2000 parameter samples from the posterior distribution. The red line is the starting value of $R_e = 1$ (situation November 2020). The green line indicates the value of $R_e$ achieved when schools for a given age group close.

society in general we assumed that (1) the number of contacts increased after April 2020 (full lockdown) but was lower than before the pandemic, and that (2) reduction in probability of transmission per contact due to mask wearing and hygiene measures was lower in August as compared to April (due to decreased adherence to measures). The starting point of our analyses is an effective reproduction number of 1.31 (95% CrI 1.15–2.07) in accordance with the state of the Dutch pandemic in August 2020 (Supplementary Fig. 4c).

Figure 5a demonstrates that in August 2020 other contacts in society in general would have to be reduced by at about 60% to bring the effective reproduction number to 1 (if school-related contacts do not change). A 100% reduction would resemble the number of contacts in April (full lockdown) and would bring the effective reproduction number to 0.83 (95% CrI 0.75–1.10). Figure 5b demonstrates that reductions of school contacts would have a limited impact on the effective reproduction number (if non-school contacts do not change). A 100% reduction (complete closure of schools) would have reduced the effective reproduction number by only 10% from 1.31 to 1.18 (95% CrI 1.04–1.83).

Subsequently, we considered the Dutch pandemic situation in November 2020 (Fig. 6), when the measures implemented since the end of August (partial lockdown intended to prevent the second wave) had led to an effective reproduction number of 1.00 (95% CrI 0.94–1.33) (Supplementary Fig. 4d). As described above, only limited control measures were taken in schools during this period. Now, the impact of interventions targeted at reducing school contacts (Fig. 6b) would reduce the effective reproduction number similarly as reducing non-school contacts in the rest of the population (Fig. 6a). Specifically, closing schools would reduce the effective reproduction number by 16% from 1.0 to 0.84 (95% CrI 0.81–0.90) (Fig. 6b). Almost the same $R_e = 0.83$ (95% CrI 0.75–1.10) would have been achieved by reducing non-school-related contacts to the level of April 2020 while the schools remain open (Fig. 6a).

**Interventions for different school ages**. Next we investigated the impact of targeting interventions at different age groups, starting from the situation in November 2020 with the effective reproduction number being about 1 (Supplementary Fig. 4d). Figure 7a–c shows $R_e$ as a function of the reduction of school contacts in age groups of [0, 5), [5, 10), and [10, 20) y.o., respectively. In each panel, we varied the number of school

contacts in one age group while keeping the number of school contacts in the other two age groups constant. Zero percent reduction corresponds to the situation in November 2020, and 100% reduction represents a scenario with schools for students in a given age group closed. The model predicts a maximum impact on $R_e$ from reducing contacts of 10–20-year-old children (Fig. 7c). Closing schools for this age group only could decrease $R_e$ by about 8% (compare Fig. 7c and Fig. 6b where we expect the reduction of 16% after closing schools for all ages). School closure for children aged 5–10 years would reduce $R_e$ by about 5% (Fig. 7b). Contact reductions among 0–5-year-old children would have a negligible impact on $R_e$ as shown in Fig. 7a.

## Discussion

We used an age-structured model for SARS-CoV-2 fitted to hospital admission and seroprevalence data during spring 2020 to estimate the impact of school contacts on transmission of SARS-CoV-2 and to assess the effects of school-based measures, including school closure, to mitigate the second wave in the autumn of 2020. We demonstrate how the relative impact of school-based measures aimed at reduction of contacts at schools on the effective reproduction number increases when the effects of non-school-based measures appear to be insufficient. These findings underscore the dilemma for policymakers of choosing between stronger enforcement of population-wide measures to reduce contacts in society in general or measures that reduce school-based contacts, including complete closure of schools. For the latter choice, our model predicts highest impact from measures implemented for the oldest school ages. We used the Netherlands as a case example but our model code is freely available and can be readily adapted to other countries given the availability of hospitalization and seroprevalence data. The findings in our manuscript can be relevant for guiding policy decisions in the Netherlands, but also in countries where the contact structure in the population is similar to that of the Netherlands[7].

Our model integrates prior knowledge of epidemiological parameters and the quantitative assessment of the model uncertainties in a Bayesian framework. To our knowledge, our modeling study is the first that uses this method to address the role of school-based contacts in the transmission of SARS-CoV-2. Previous studies (e.g. refs. [21–25]) used individual-based or network models that were not fit to epidemiological data using formal statistical procedures. Due to uncertainties in key model

parameters, predictions of these models vary widely. Our model has been carefully validated to achieve an excellent fit to data of daily hospitalizations due to COVID-19 and seroprevalence by age. Furthermore, reproduction numbers at different time points of the pandemic correlated well with estimates obtained from independent sources[1]. In addition, the Netherlands is one of few countries in the world for which the contact rate after the first lockdown is available (see ref. [26] for the UK). We could, therefore, model the contact structure during the course of the pandemic as continuously changing between the contact structure before the pandemic and the contact structure when the measures were the most strict (first lockdown) without making additional assumptions about the impact of specific interventions on contacts in different age groups. Crucially, many prior models evaluating the impact of school-based contacts assumed age-independent susceptibility to infection with SARS-CoV-2 (e.g. refs. [13,23]). Here, we estimated susceptibility to infection with SARS-CoV-2 to increase with age, which corroborates published findings from cohort studies[8,9]. Compared to adults older than 60 years, the estimated susceptibility was about 20% for children aged 0–20 years and about 60% for the age group of 20–60 years. However, even with extensive validation, we need to be careful when interpreting the predictions of our model as these depend on the sensitivity of serology to identify individuals with prior infection. Recent studies suggest that in persons who experience mild or asymptomatic infections, SARS-CoV-2 antibodies may not always be detectable post-infection[27,28]. Therefore, more children may have had an infection than indicated by the seroprevalence survey because the proportion of asymptomatic in children is believed to be high. As a consequence, our study potentially underestimates the role of children in transmission.

Naturally, our findings result from age-related differences in disease susceptibility and contact structure. Despite high numbers of contacts for children of all ages, and in particular in the age group of 10–20 years old, closing schools appeared to have much less impact on the effective reproduction number than contact-reduction measures outside the school environment. In fact, measures effectively reducing non-school contacts, similar to those measures implemented in response to the first pandemic wave in spring 2020, could have prevented a second wave in autumn without school closures. With an estimated effective reproduction number of 1.3 in August 2020, continuation of school closures would have had much lower effects than measures aiming to reduce non-school-related contacts, which mainly occur in the adult population. Yet, that situation changes if the proposed measures fail. In November 2020, the measures implemented since August had reduced the effective reproduction number to around 1, instead of achieving the target value of about 0.8. In that situation, as our findings demonstrate, additional physical distancing measures in schools could assist in reducing the effective reproduction number further, in particular when implemented in secondary schools. Our analyses suggest that physical distancing measures in the youngest children will have no impact on the control of SARS-CoV-2 infection. Of note, better adherence to non-school-based measures would still have similar effects as reducing school-based contacts.

Although there are several options for reducing the number of contacts between children at school, such as staggered start and end times and breaks, different forms of physical distancing for pupils and division of classes, the effects of such measures on transmission among children have not been quantified. Importantly, we have assumed that reductions in school-based contacts are not replaced by non-school-based contacts (among children and between children and adults) with similar transmission risk.

Our modeling approach has several limitations. For estimating disease susceptibility we could only model children as a group of 0–20 years old. As disease susceptibility increases with age, it seems obvious that effects of reduced school contacts are most prominent in older children. Assuming equal susceptibility across these ages may have underestimated to some extent the effect of reducing school contacts for children between 10 and 20 years. At the same time, we assumed that school contact patterns in August–November 2020 reflect the pre-pandemic situation. Yet, general control measures in the Netherlands such as stay at home orders for symptomatic persons probably lower infectious contacts in school settings too, meaning that some reduction compared to pre-pandemic levels of contacts could already be present in schools. Effects of these measures in school settings should be smaller than in the general population and are hard to estimate due to a large number of asymptomatic cases among children, and therefore were not taken into account. In this respect, the results reported here describe the maximum possible reduction in the effective reproduction number due to school interventions. Furthermore, the contact matrices available did not allow differentiation between various types of contacts outside schools (like work, leisure, transport, etc.), as these were not available for periods during the pandemic. Therefore, we could not model the impact of reducing work-related or leisure-related contacts separately. We also could not include hospitalization data from the second wave of the pandemic due to lack of data availability.

The potential effects of opening or closing schools in different phases of the pandemic have been reported in other studies[13,21–24,29,30]. Also based on a mathematical model, Panovska-Griffiths et al.[21] predicted that without very high levels of testing and contact tracing reopening schools after summer with a simultaneous relaxation of measures will lead to a second wave in the United Kingdom, peaking in December 2020. Their model predicted that this peak could be postponed for two months (to February 2021) by a rotating timetable in schools. Very early in the pandemic, in March 2020, the Scientific Advisory Group for Emergencies in the United Kingdom concluded that it would not be possible to reduce the effective reproduction number below 1 without closing schools[29]. In a modeling study on the impact of non-pharmaceutical interventions for COVID-19 in the United Kingdom, Davies et al.[13] found that the impact of school closures was low. In another modeling study Rice et al.[24] found that school closures during the first wave of the pandemic could increase overall mortality, due to death being postponed to a second and subsequent waves. And based on an analysis of the impact of non-pharmaceutical measures in 41 countries between January and May 2020, Brauner et al.[30] concluded that closure of schools and universities had contributed the most to lowering the effective reproduction number. Yet, a major difficulty in estimating the effect of school closure using observational data from the first wave is that other non-pharmaceutical interventions were implemented at or around the same time as school closures[31]. Similarly, lifting such measures often coincided with school re-openings. Observational data from the period after the first wave show conflicting results on within school transmission[32–35] and the effect of school reopening and interpretation is further hampered by the variety in control measures implemented in schools across countries. Finally, Munday et al.[22] showed that reopening secondary schools is likely to have a greater impact on community transmission than reopening primary schools in England. While the modeling approach of Munday et al.[22] is different from ours, our findings are similar in the sense that secondary schools are predicted to make a larger contribution to transmission than primary schools, and are therefore more important for controlling COVID-19.

In conclusion, we have demonstrated that the potential effects of school-based measures to reduce contacts between children, including school closures, markedly depends on the reduction in

the effective reproduction number achieved by other measures. With remaining opportunities to reduce the effective reproduction number with non-school-based measures, the additional benefit of school-based measures is low. Yet, if opportunities to reduce the effective reproduction number with non-school-based measures are considered to be exhausted or undesired for economic reasons and $R_e$ is still close to 1, the additional benefit of school-based measures may be considerable. In such situations, the biggest impact on transmission is achieved by reducing contacts in secondary schools.

## Methods

**Overview**. Estimates of epidemiological parameters were obtained by fitting a transmission model to age-stratified COVID-19 hospital admission data in the period from 27 February till 30 April 2020 ($n = 10{,}961$) and cross-sectional age-stratified SARS-CoV-2 seroprevalence data assessed in April/May 2020 ($n = 3207$)[18]. The model equipped with parameter estimates was subsequently used to investigate the impact of school- and non-school-based measures on controlling the pandemic.

**Data**. The hospital data included $n = 10{,}961$ COVID-19 hospitalizations by date of admission and stratified by age during the period of 64 days following the first official case in the Netherlands (27 February 2020). The criteria for hospital admission have not changed during the pandemic, and from the early stages all hospitalized patients with a clinical suspicion of COVID-19 were tested by RT-PCR. In all stages of the pandemic, patients requiring hospital admission were hospitalized and the practice of not referring patients for hospital admission (e.g. due to self-expressed treatment restrictions or moribund condition) did not change.

The SARS-CoV-2 seroprevalence data were taken from a cross-sectional population-based serological study carried out in April/May 2020 (PIENTER Corona study)[18]. Participants for the serosurvey were enrolled from a previously established nationwide serosurveillance study, provided a self-collected fingerstick blood sample and completed a questionnaire. A total of 40 municipalities were randomly selected, with probabilities proportional to their population size. From these municipalities, an age-stratified sample was drawn from the population register, and 6102 persons were invited to participate. Serum samples and questionnaires were obtained from 3207 participants and included in the analyses. The majority of blood samples were drawn in the first week of April. IgG antibodies targeted against the spike S1-protein of SARS-CoV-2 were quantified using a validated multiplex immunoassay. Seroprevalence was estimated controlling for survey design, individual pre-pandemic concentration, and test performance.

Our analyses made use of the demographic composition of the Dutch population in July 2020 from Statistics Netherlands[36] and age-stratified contact data for the Netherlands[19,20]. The contact rates before the pandemic were based on a cross-sectional survey carried out in 2016/2017, where participants reported the number and age of their contacts during the previous day[19]. The contact rates after the first lockdown were based on the same survey which was repeated in a sub-sample of the participants in April 2020 (PIENTER Corona study)[19]. School-specific contact rates for the Dutch population before the pandemic were taken from the POLYMOD study[7,20].

**Transmission model**. We used a deterministic compartmental model describing SARS-CoV-2 transmission in the population of the Netherlands stratified by infection status and age (Fig. 8a). Some modeling studies on the impact of interventions against COVID-19 account for spatial variations of the disease[16,17,37]. Since the available data are aggregated on the country level and for the sake of the model's tractability, we disregarded regional stratification of the population. The dynamics of the model follows the Susceptible-Exposed-Infectious-Recovered structure. Persons in age group $k$, where $k = 1, \ldots, n$, are classified as susceptible ($S_k$), infected but not yet infectious (exposed, $E_k$), infectious in $m$ stages ($I_{k,p}$, where $p = 1, \ldots, m$), hospitalized ($H_k$), and recovered without hospitalization ($R_k$). Susceptible persons ($S_k$) can acquire infection via contact with infectious persons ($I_{k,p}$, $k = 1, \ldots, n$, $p = 1, \ldots, m$) and become latently infected ($E_k$) at a rate $\beta_k \lambda_k$, where $\lambda_k$ is the force of infection, and $\beta_k$ is the reduction in susceptibility to infection of persons in age group $k$ compared to persons in age group $n$. Persons in the classes $I_{k,p}$, ($k = 1, \ldots, n$, $p = 1, \ldots, m$) are assumed to be equally infectious. After the latent period (duration $1/\alpha$ days), exposed persons become infectious ($I_{k,1}$). Infectious persons progress through ($m - 1$) stages of infection ($I_{k,p}$, where $p = 2, \ldots, m$) at rate $\gamma m$, after which they recover ($R_k$). Inclusion of $m$ identical infectious stages allows for the tuning of the distribution of the infectious period (the time spent in the infectious compartments, $I_{k,p}$, $p = 1, \ldots, m$)[38,39], interpolating between an exponentially distributed infectious period ($m = 1$) and a fixed infectious period ($m \to \infty$). Intermediate values of $m$ correspond to an Erlang-distributed infectious period with mean $1/\gamma$ and standard deviation $1/[\gamma\sqrt{m}]$. Hospitalization ($H_k$) of infectious persons ($I_{k,p}$) occurs at rate $\nu_k$. Since the model is fitted to hospital admissions data, the disease-related mortality and discharge from the hospital are not explicitly modeled, and $H_k$ describes the cumulative number of

hospital admissions. We assume that currently hospitalized persons (who may still be infectious) will have contacts with medical personnel and visitors, but these persons will not be infected because they use personal protective measures. Given the timescale of the pandemic and the lack of reliable data on reinfections, we assume that recovered individuals cannot be reinfected. As the timescale of the pandemic is short compared to the average lifespan of persons, we neglected natural birth and death processes, and the population size in the model stays constant.

We assume that, before the first lockdown, the probability of transmission per contact between a susceptible and an infectious individual, $\epsilon$, is independent of the age of two individuals. After introduction of the control measures in March 2020, this probability of transmission decreased to $\epsilon\zeta_1$, where $0 \le \zeta_1 \le 1$. The value $(1 - \zeta_1)$ then denotes the reduction in the probability of transmission due to general population-based measures that are not explicitly included in the model, such as refraining from shaking hands, mask wearing, and self-isolation of symptomatic persons.

We denote the general contact rate (the number of contacts per day) of a person in age group $k$ with persons in age group $l$, $c_{kl}(t)$, and the contact rates specific to the periods before and after the first lockdown, $b_{kl}$ and, $a_{kl}$, respectively (see Fig. 8b, c). The contacts are defined as contacts with household members and contacts in the community[19]. Examples of a contact outside one's household are talking to someone (face-to-face), touching someone, kissing someone, or doing sports with someone. More details on contact matrices $a_{kl}$ and $b_{kl}$ can be found in ref. [19]. We assume a smooth change from the contact rate $b_{kl}$ to the contact rate $a_{kl}$, as the contact-reduction measures were introduced during the first lockdown. We model the transition in the general contact rate from before to after the first lockdown using a linear combination

$$c_{kl}(t) = [1 - f(t)]b_{kl} + \zeta_1 f(t)a_{kl}, \tag{1}$$

where the contribution of the contact rate after the first lockdown is given by the logistic function

$$f(t) = \frac{1}{1 + e^{-K_1(t - t_1)}} \tag{2}$$

with the mid-point value $t_1$ and the logistic growth $K_1$ (Supplementary Fig. 1). The parameter $K_1$ governs the speed at which control measures are rolled out, and $t_1$ is the mid-time point of the lockdown period. The special cases of $f = 0$ and $f = 1$ describe the contact rate before and after the first lockdown, with $f$ values between 0 and 1 corresponding to contact rates at the intermediate time points.

To investigate the impact of school- and non-school-based measures individually, we need to be able to split the contact rate into a rate of contacts occurring at schools and a rate of contacts occurring elsewhere. The contact rates we used from the literature are additive[19,20]; thus, the contact rate before the lockdown ($b_{kl}$) can be written as a sum of the school contact rate at the pre-lockdown level ($s_{kl}$, see Fig. 8d) and the contact rate for all locations but schools ($b_{kl} - s_{kl}$). The contact rate after the lockdown ($a_{kl}$) by definition did not include any school contacts because all schools were closed. The contact rate incorporating the relaxation of control measures after the first lockdown is therefore modeled as follows:

$$c_{kl}(t) = \zeta_1 g(t)a_{kl} + [1 - g(t)]\zeta_2(b_{kl} - s_{kl}) + \omega s_{kl}, \tag{3}$$

where $g(t) = 1/\left[1 + e^{K_2(t - t_2)}\right]$ with the mid-point value $t_2 > t_1$ and the logistic growth $K_2$. In Eq. (3), the first two terms describe the increase of non-school contacts from the level after the first lockdown ($a_{kl}$) to their pre-lockdown level ($b_{kl} - s_{kl}$). The parameter $\zeta_2 \ge \zeta_1$, $0 \le \zeta_2 \le 1$ implies that the probability of transmission increased due to reduced adherence to control measures. The last term describes opening of schools which we assume to happen instantaneously, where $\omega$, $0 \le \omega \le 1$, is the proportion of retained school contacts. Schools functioning without any measures correspond to $\omega = 1$. School closure is achieved by setting $\omega = 0$. A summary of the model parameters is given in Table 1.

**Model equations**. The model was implemented in Mathematica 10.0.2.0 using a system of ordinary differential equations as follows:

$$\begin{aligned}
\frac{dS_k(t)}{dt} &= -\beta_k \lambda_k(t) S_k(t), \\
\frac{dE_k(t)}{dt} &= \beta_k \lambda_k(t) S_k(t) - \alpha E_k(t), \\
\frac{dI_{k,1}(t)}{dt} &= \alpha E_k(t) - (\gamma m + \nu_k) I_{k,1}(t), \\
\frac{dI_{k,p}(t)}{dt} &= \gamma m I_{k,p-1}(t) - (\gamma m + \nu_k) I_{k,p}(t), \qquad p = 2, \ldots, m, \\
\frac{dR_k(t)}{dt} &= \gamma m I_{k,m}(t), \\
\frac{dH_k(t)}{dt} &= \nu_k \sum_{p=1}^{m} I_{k,p}(t),
\end{aligned} \tag{4}$$

where $S_k$, $E_k$, $R_k$, and $H_k$ are the numbers of persons in age group $k$, $k = 1, \ldots, n$, who are susceptible, exposed, recovered, and hospitalized, respectively. The number of infectious persons in age group $k$ and stage $p = 1, \ldots, m$ is denoted $I_{k,p}$. The

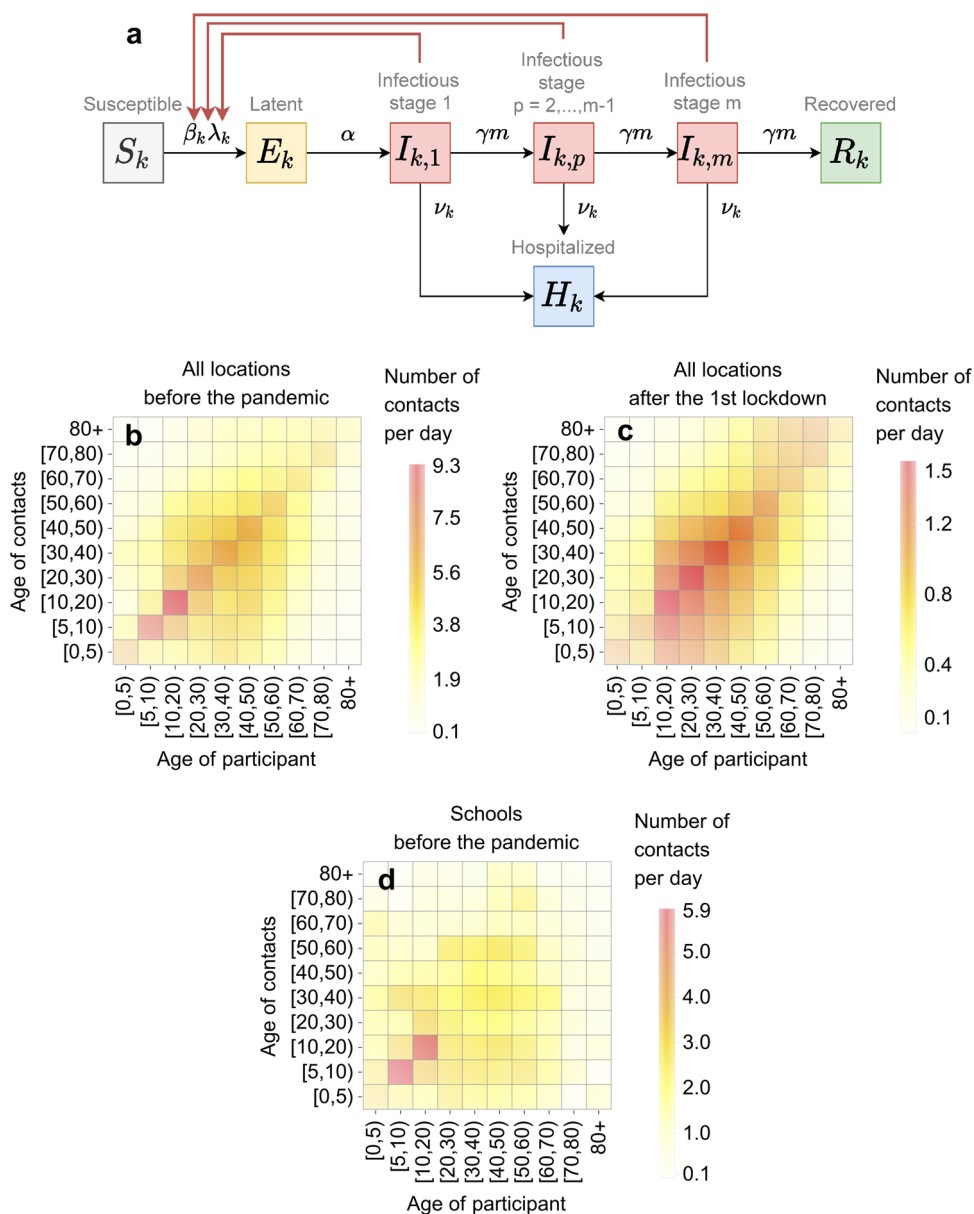

**Fig. 8 Transmission model and contact rates. a** Model schematic. Black arrows show epidemiological transitions. Red arrows indicate the compartments contributing to the force of infection. Susceptible persons in age group $k$ ($S_k$), where $k = 1, ..., n$, become latently infected ($E_k$) via contact with infectious persons in $m$ infectious stages ($I_{k,p}$, $p = 1, ..., m$) at a rate $\beta_k \lambda_k$, where $\lambda_k$ is the force of infection, and $\beta_k$ is the reduction in susceptibility to infection of persons in age group $k$ compared to persons in age group $n$. Exposed persons ($E_k$) become infectious ($I_{k,1}$) at rate $\alpha$. Infectious persons progress through ($m - 1$) infectious stages at rate $\gamma m$, after which they recover ($R_k$). From each stage, infectious persons are hospitalized at rate $\nu_k$. Table 1 gives the summary of the model parameters. **b–d** Contact rates. **b**, **c** show contact rates in all locations before the pandemic and after the first lockdown (April 2020), respectively; **d** shows contact rates at schools before the pandemic. The color represents the average number of contacts per day a person in a given age group had with persons in another age group.

force of infection is given by

$$\lambda_k(t) = \epsilon \sum_{l=1}^{n} \sum_{p=1}^{m} c_{kl}(t) \frac{I_{l,p}(t)}{N_l}, \tag{5}$$

where $N_l$ is the number of individuals in age group $l$,

$N_l = S_l(t) + E_l(t) + \sum_{p=1}^{m} I_{l,p}(t) + H_l(t) + R_l(t)$. Note that the denominator in the

force of infection ($N_l$) includes hospitalized persons, $H_l(t)$, where $H_l(t)$ describes the cumulative number of hospital admissions at time $t$. The current number of hospitalized persons (not the cumulative) is not subtracted from $N_l$ because we assume that hospitalized persons will be involved in contacts with medical personnel and visitors. Since we assume that contacts of the currently hospitalized persons (who may still be infectious) will not be infected due to the use of personal protective measures by medical personnel and hospital visitors, the current number

of hospitalized persons does not contribute to the force of infection. As patients who are discharged and recovered (or deceased) also do not contribute to the force of infection, the cumulative number of hospitalized persons ($H_l(t)$) does not contribute to the force of infection either. In Eq. (5) we assumed a frequency-dependent transmission where the per capita rate at which a susceptible person becomes infected increases with the fraction of the population that is infectious. This choice is justified for the Netherlands as one of the most densely populated countries in Europe. Moreover, as the population size does not change during the time horizon of our analyses, there is no difference in the outcome between a frequency-dependent and a density dependent model once the parameters are fitted to obtain the observed reproduction number.

We took 22 February 2020 as starting date ($t_0$) for the pandemic in the Netherlands, which is 5 days prior to the first officially notified case (27 February 2020). We assumed that there were no hospitalizations during this 5-day period. To account for importation of new cases into the Netherlands at the beginning of the pandemic, we

**Table 1 Summary of the model parameters.**

| Description (unit) | Notation | Reference |
|---|---|---|
| *Constant parameters* | | |
| Number of age groups | $n$ | 10 |
| Number of infectious stages | $m$ | 3 |
| Basic reproduction number | $R_0$ | Computed using the method in ref. [45] |
| Effective reproduction number | $R_e$ | Computed using the method in ref. [45] |
| Probability of transmission per contact | $\epsilon$ | Estimated |
| Reduction in post-lockdown probability of transmission per contact | $(1 - \zeta_1)$ | Estimated |
| Latent period (days) | $1/\alpha$ | Estimated |
| Infectious period (1/day) | $1/\gamma$ | Estimated |
| Contribution of the contact rate after the lockdown | $f(t) = 1/\left[1 + e^{-K_1(t-t_1)}\right]$ | Eq. (2) |
| Mid-point value of the logistic function (days) | $t_1$ | Estimated |
| Logistic growth (1/day) | $K_1$ | Estimated |
| Over-dispersion parameter for the NegBinom distribution for hospitalizations | $r$ | Estimated |
| Proportion of school contacts | $\omega$ | [0, 1], calibrated |
| Reduction in probability of transmission per contact during relaxation | $(1 - \zeta_2)$ | [0, 1], $\zeta_2 \geq \zeta_1$, calibrated |
| Initial fraction of infected persons | $\theta$ | Estimated |
| Logistic function for relaxation | $g(t) = 1/\left[1 + e^{K_2(t-t_2)}\right]$ | $0 \leq g(t) \leq 1$, calibrated |
| *Age-specific parameters*[a] | | |
| Force of infection (1/day) | $\lambda_k$ | Eq. (5) |
| Hospitalization rate (1/day) | $\nu_k$ | Estimated |
| Susceptibility of age group $k$ relative to age group $n$[b] | $\beta_k$ | Estimated |
| General contact rate (1/day) | $c_{kl}$ | Eqs. (1) and (3) |
| Contact rate before the pandemic (1/day) | $b_{kl}$ | 19 |
| Contact rate after the first lockdown (1/day) | $a_{kl}$ | 19 |
| School contact rate before the pandemic (1/day) | $s_{kl}$ | 7,20 |
| Population size of age group $k$ | $N_k$ | 36 |

[a]Indices $k$ and $l$ denote the age groups $k, l = 1, ..., n$.
[b]In the estimation procedure the reference age group $n$ is 60+, and $\beta_{60+} = 1$ is fixed at 1.

estimated a fraction $\theta$ of each age group infected at time $t_0$. For simplicity, we assumed this fraction to be equally distributed between the exposed and infectious persons, i.e., $E_k(t_0) = \frac{1}{2}\theta N_k$, $I_{k,p}(t_0) = \frac{1}{2m}\theta N_k$ and $S_k(t_0) = (1-\theta)N_k$. In later stages of the pandemic, importations do not play such an important role because of existing pool of infected individuals within the country and ongoing control measures. For this reason, importations after $t_0$ were not included in the model.

**Observation model and parameter estimation.** Given predictions of the model, we calculated the likelihood of the data as follows. In the model, infectious individuals are hospitalized at a continuous rate $\nu_k \sum_{p=1}^{m} I_{k,p}$. However, the hospitalization data consist of a discrete number of hospital admissions $h_{k,i}$ on day $T_i$ for each age class $k$. As the probability of hospitalization is relatively small, we made the simplifying assumption that the daily incidence of hospitalizations is proportional to the prevalence of infectious individuals at that time point. To accommodate errors in reporting and within age class variability of the hospitalization rate, we allowed for over-dispersion in the number of hospitalizations using a Negative-Binomial distribution, i.e.,

$$h_{k,i} \sim \text{NegBinom}\left(\nu_k \sum_{p=1}^{m} I_{k,p}(T_i), r\right), \quad (6)$$

where we parameterize the NegBinom$(\mu, r)$ distribution with the mean $\mu$ and over-dispersion parameter $r$, such that the variance is equal to $\mu + \mu^2/r$.

We calculated the likelihood of the seroprevalence data using the model prediction of the fraction of non-susceptible individuals in each age class $1 - S_k(T)/N_k$. Here $T$ denotes the median sampling time minus the expected duration from infection to seroconversion. We assumed that the probability of finding a seropositive individual in a random sample from the population is equal to the fraction of non-susceptible individuals, leading to a Binomial distribution for the number of positive samples $\ell_k$ among all samples $L_k$ from age group $k$

$$\ell_k \sim \text{Binom}(L_k, 1 - S_k(T)/N_k). \quad (7)$$

Parameters were estimated in a Bayesian framework based on the methods from refs. [40,41]. The model given by Eq. (4) was fit to the data using the Hamiltonian Monte Carlo method as implemented in Stan (https://www.mc-stan.org)[42] with R and R Studio interfaces. We used four parallel chains of length 1500 with a warm-up phase of length 1000, resulting in 2000 parameter samples from the posterior distribution.

We used age-specific contact rates with ten age groups (ages [0, 5), [5, 10), [10, 20), [20, 30), [30, 40), [40, 50), [50, 60), [60, 70), [70, 80) and 80+). Due to the low number of hospitalizations in young persons, we assumed that hospitalization rates in the first three age groups (ages [0, 5), [5, 10), [10, 20)) were equal; therefore, only eight

hospitalization rates were estimated. As the age-specific hospitalization rates are positively correlated (Supplementary Fig. 5), we parameterized the model as $\nu_k = \hat{\nu}_k \bar{\nu}$, where $\hat{\nu}_k$ is a simplex and $\bar{\nu}$ a scalar. We kept the same age categories for the relative susceptibility as in the retrospective cohort study by Jing et al.[8], from where we took the priors, i.e., the relative susceptibility was estimated for ages [0, 20), [20, 60), and 60+ age category was used as the reference corresponding to susceptibility equal to 1. As the age groups for which the seroprevalence was reported[18] are different from the age groups used in our model, we used demographic data from the Netherlands[36] and the smoothed age-specific seroprevalence curve estimated by Vos et al.[18] to correct for this discrepancy. The Bayesian prior distributions for the 10 estimated parameters (18 numbers in total as hospitalization rate and susceptibility are age-dependent) (see Table 1) are listed in Table 2. In the main text, we presented results for three infectious classes corresponding to an Erlang-distributed infectious period with shape parameter $m = 3$.

**Model outcomes.** We considered control measures aimed at reducing contact rate at schools or in all other locations. We evaluated the impact of a control measure by computing $R_e$ using the next-generation matrix (NGM) method[43–46], and percentage of contacts that need to be reduced to achieve control of the pandemic as quantified by $R_e = 1$. Previously, we applied this method for HIV and CMV transmission models[41,47]. The method for calculating the basic reproduction number $R_0$ and $R_e$ (Supplementary Fig. 4) is described in detail in Supplementary Information. In short, Supplementary Fig. 4a, b were obtained using the NGM method and posterior distributions of the parameters (Supplementary Fig. 3) that were estimated from fitting the model to the data of the first 69 days of the pandemic (22 February till 30 April 2020). As hospitalization data during relaxation are not available, we calibrated the model to values of $R_e$ as published on the dashboard of the National Institute for Public Health and the Environment (RIVM)[1]. These time-dependent $R_e$ values are estimated from hospitalization data and later from case numbers using methods described in ref. [48]. Specifically, we chose $\omega$, $g$, and $\zeta_2$ such that the median reproduction numbers in the model would equal the specific values estimated by the RIVM (about 1.3 in the period 27 August–6 September and about 1 in the period 7–13 November)[1]. The distributions shown in Supplementary Fig. 4c, d are therefore obtained using the NGM method with fixed $\omega$, $g$, and $\zeta_2$ and other parameters drawn from the posterior distributions as shown in Supplementary Fig. 3. Note that the calibration of the model in the relaxation period is possible because the parameters describing epidemiology of SARS-CoV-2 are assumed to be constant throughout the time horizon of the analyses which spans both pre-lockdown, post-lockdown, and relaxation periods, and only contact structure varies with time (see Supplementary Information). In analyses, the parameters $\omega$ and $g$ were then used as control parameters to reduce the number of school- and non-school-

**Table 2 Prior distributions for the Bayesian statistical model.**

| Parameter | Prior[a] | Explanation |
|---|---|---|
| $\epsilon$ | Uniform $(0, 1)$ | Flat prior |
| $\alpha$ | InvGamma $(32.25, 9.75)$ | 99% of the prior density of $1/\alpha$ between 2 and 5 days |
| $\gamma$ | InvGamma $(22.6, 2.44)$ | 99% of the prior density of $1/\gamma$ between 5 and 15 days |
| $\nu_{[0,20)}, \nu_{[20,30)}$ $\nu_{[30,40)}, \nu_{[40,50)}$ $\nu_{[50,60)}, \nu_{[60,70)}$ $\nu_{[70,80)}, \nu_{80+}$ | folded-$\mathcal{N}(0,5)$ | Vague prior |
| $\beta_{[0,20)}$[b] | Lognormal $(-1.47, 0.1)$ | Log-odds $-1.47 = \log(0.23)$ based on prior estimates[8] |
| $\beta_{[20,60)}$[b] | Lognormal $(-0.45, 0.1)$ | Log-odds $-0.45 = \log(0.64)$ based on prior estimates[8] |
| $r$ | Lognormal$(5, 2)$ | Vague prior |
| $\zeta_1$ | $\mathcal{N}(1, 0.1)$ | A priori, we expect the reduction in contacts after the first lockdown to account for most of the decrease in the transmission rate |
| $t_1$ | $\mathcal{N}(23, 7)$ | The mean of $t_1$ is given by the day of initiation of most drastic social distancing measures (15 March); most measures were taken within two weeks from this date |
| $K_1$ | Exp$(1)$ | For $K_1 = 1$ the uptake of measures takes approximately 6 days |
| $\theta$ | Uniform $(10^{-7}, 5 \times 10^{-4})$ | Vague prior allowing for approximately $10^0$–$10^5$ infections at time $t_0$ |

[a]The scale parameter of the normal and log-normal distributions is equal to the standard deviation.
[b]$\beta_{60+} = 1$ for the reference group of 60+.

related contacts (Figs. 5–7). In doing so, we varied one type of contact and kept the other type constant.

**Reporting summary**. Further information on research design is available in the Nature Research Reporting Summary linked to this article.

## Data availability

All datasets analyzed and generated during this study are available in the GitHub repository, https://github.com/lynxgav/COVID19-schools[49].

## Code availability

Mathematica, Stan, R, and R Studio codes reproducing the results of this study are available in the GitHub repository, https://github.com/lynxgav/COVID19-schools[49].

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

## Acknowledgements

The contribution of C.H.v.D. was under the auspices of the US Department of Energy (contract number 89233218CNA000001) and supported by the National Institutes of Health (grant number R01-OD011095). M.E.K. was supported by ZonMw grant number 10430022010001, ZonMw grant number 91216062, and H2020 project 101003480 (CORESMA). M.J.M.B. and P.B.-V. were supported by H2020 project 101003589 (RECOVER). G.R. was supported by FCT project 131_596787873. We thank Michiel van Boven (The National Institute of Public Health and the Environment, Bilthoven, The Netherlands) and Ana Nunes (Lisbon University) for valuable discussions and continuing advice during the course of this project. We thank João Viana for validating the Mathematica code. We thank Mui Pham and Alexandra Teslya for comments on the manuscript. We thank Eric Vos and Jantien Backer for the information on the serological and contact data used in this study.

## Author contributions

G.R. conceived the study and drafted the first version of the manuscript. G.R. and C.H.v.D. developed the model with input from M.E.K., M.B. and P.B.-V. Statistical inference and sensitivity analyses were performed by G.R. and C.H.v.D. G.R. implemented control measures, carried out all model analyses, and prepared figures. M.E.K., M.C.J.B., and J.H.H.M.v.d.W. validated the model and analyses. All authors contributed to analysis, interpretation of the results, writing the final version of the manuscript, and gave final approval for publication.

## Competing interests

The authors declare no competing interests.
