## [Peer Review File · Nature Communications]

Reviewers' Comments:

Reviewer #1:

Remarks to the Author:

Review of “Model-based evaluation of school- and non-school-related measures to control the COVID-19 pandemic” by Ganna Rozhnova et al.

This ms deals with the problem of assessing the role played by school-related containment measures in curbing the COVID-19 epidemic in the Netherlands. This is achieved by implementing and calibrating a model that includes different age classes and different disease stages. Reproduction numbers are estimated via a Next Generation Matrix approach. This approach was also taken e.g. by Gatto et al. (*PNAS*, 2020, 117 (19) 10484-10491), but the categories were different locations in Italy, not different age classes. Calibration is performed by fitting data of hospitalizations and seroprevalence, a good choice. The ms is very interesting, as it faces an important issue, that of keeping schools closed or at least partially open. It relies on sound data and the authors are clearly competent in the art of epidemiological modelling. However there are some points of the ms that do require clarification or further analysis.

- The first point concerns the very estimation of the effective reproduction number. The only hint at how R_e is estimated is given at line 172, where the authors state “we evaluated the impact of a control measure by computing the effective reproduction number (R_e) using the next generation matrix method [18, 22].” However, both ref. 18 and 22 specify how to compute R_0 , the basic reproduction number, not the effective reproduction number, if by effective we mean that number at times later than 0, when the prevalence of susceptibles is no longer 1 (Disease-free equilibrium) and the value of the epidemiological parameters is no longer that prevailing at the beginning of the epidemic. Control measures can be accounted for according to 18 and 22, but the calculated R_e is nothing but R_0 in which the parameter values are instantaneously changed at the very beginning when the prevalence of susceptibles is 1. This describes the situation in which the control measures are instantaneously and immediately deployed. I suspect, instead, that the authors have calculated a time-varying next generation matrix which includes both the instantaneous prevalence of susceptibles (<1) and the time-varying values of the epidemiological parameters. Then the spectral radius of the NGM has been calculated at any time. If this is indeed the procedure, I can sympathize with it, even if it is based on practical more than theoretical grounds. However, the authors should a) describe their procedure exactly, b) provide some sort of justification to other literature.
- The model has been calibrated on the data of the first 69 days after February 27. The parameters, part of which are time varying, are therefore the parameters that describe the variations along these 69 days. Therefore, how did the authors obtain estimates of the effective reproduction number after May 6 (fig. 5)? It is not only a matter of procedure (point above) but also a problem of how the various epidemiological parameters have been estimated, given that the data on hospitalizations and serology refer to the initial period only (line 63 and following). Calculating the time-varying NGM requires that the variations of say the contact rates are estimated. Thus, it is not at all clear how R_e is estimated after May 6 and how its distribution is obtained (Fig. S4C-D). At lines 214-215 the authors claim that $R_e = 1.31$ should be achieved. I do not understand why such a value should be “achieved”. Where does 1.31 come from? How was it estimated? Similarly, at line 226 the authors state that in November 2020 (not clear which day) an effective reproduction number of 1.00 had been reached. Again, where does this number come from? At line 264 the authors state that reproduction numbers at different time points of the pandemic correlated well with estimates obtained from independent sources (Dutch dashboard). Given that the model was fit to data in the first 69 days, how were these numbers at different times obtained? Purportedly, by using methods and/or

data different from those of the Dutch dashboard. But where are the methods and the data described?

- As a whole, 10 parameters (if I am not wrong, see Table 1) have been estimated. Actually, two of these parameters depend on age (hospitalization rate and susceptibility). As far as I understand the estimated hospitalization rates are 8 (Fig. 3) and the susceptibilities are 3 (ages 0-20, 20-60, 60+). A total of 19 values to be estimated, consequently. Since this number is quite large, it would be interesting to know whether there are correlations (negative or positive) between parameters. If it were so, this would suggest that a more parsimonious model might have been used. For instance, why did the authors choose exactly those 3 categories for susceptibility? I suggest that the estimated correlation matrix between the 19 parameters is reported in the Supplementary Information and commented in the main text. As the authors have implemented a Bayesian Monte Carlo method, it is no problem for them to obtain such a matrix. Some comments on the choice of the number of parameters to be estimated are also necessary.

Other comments

- The main text of the paper should be made shorter avoiding unnecessary repetitions and delegating part of the technicalities to the supplementary information.
- The authors have chosen to disregard the spatial variations of the disease, differently from e.g. Gatto et al. (2020) and Bertuzzo et al. (*Nature Communications*, **11**, 4264, 2020) for Italy. This is somehow justified by the high populations density prevailing in the Netherlands and the large connectivity of their transportation system. A comment on this choice, however, would be welcome.
- line 47: “contract” → contact
- line 86: “Inclusion of m identical infectious period etc.” Yes and no. What is Erlang- p distributed is the time between an exposed becoming infectious and a susceptible becoming infected by an individual in the class $I_{k,p}$. So, if one wants an Erlang- m distribution of the infectious period (with $m = 3$ in this ms), one should implement a model in which a susceptible is infected only by individuals in the class $I_{k,m}$. The other classes are purely fictitious and serve only the purpose of generating the desired Erlang- m distribution. If all the classes $I_{k,p}$ are assumed to be equally infectious (parameter β_k) then the infectious period is distributed as the sum of Erlang-1 plus Erlang-2 etc. I am afraid that the paper D. Champredon, J. Dushoff, D.J.D. Earn, Equivalence of the Erlang-distributed SEIR epidemic model and the renewal equation, *SIAM J. Appl. Math.* 78(2018) is frequently misunderstood. What Champredon et al. call the Erlang-distributed SEIR epidemic model does not have an Erlang distributed generation interval (as the authors actually show in their sect.3)! All this is related to hyper-Erlang distributions in queueing systems. I think that the present ms should be amended in this respect.
- Legend of Fig. 1: Authors should specify that the number of contacts is actually “number of contacts per day”.
- Line 100: Here the authors might specify that contact rate is actually measured as number of close contacts per day. I assume that the authors by contacts mean “close” contacts. Maybe they should better explain this point.
- Line 114 (eq. 3): I think that the choice of subtracting school contacts from pre-lockdown contacts is not very clear. The authors should clarify this point.
- Line 134: “ $N_k = S_k$ etc.” We cannot think that hospitalized (and quarantined, not modelled here) individuals are mobile. The correct force of infection is that reported in Gatto_et_al (2020) and Bertuzzo_et_al (2020). It probably does not make such a big difference in

practice, but conceptually the calculation of N_k is wrong. Also, the assumption of frequency-dependent force of infection, rather than e.g. density-dependent, should be commented.

- Line 157: “Parameters were estimated in a Bayesian framework using methods we developed before”. Please give the reader at least a hint about the nature of these methods.
- Legend of Fig.2: Specify the starting date (27 February day 0 ?).
- Line 322: “sunsequent” → “subsequent”

Reviewer #2:

Remarks to the Author:

This is a standard age-stratified SEIR model to assess the impact of school closure in the Netherlands. The model was parameterized based on backward inference from empirical hospitalization and seroprevalence data.

I am concerned about the treatment of the two key empirical datasets without due regard to potential biases in the way they were assembled. Time varying (depending on pandemic stage) factors that would have affected case ascertainment, identification through testing and admission threshold, in addition to laboratory details regarding the veracity/validity of serosurveys should have been explicated and taken into fuller account in the modelling.

The manuscript is written in such a way that it does not highlight the salient points of the background, methods or findings. It reads more like a detailed log.

The authors should explain how they treated importation seeding, and why or why not.

Model details should be checked and the code vetted – I have not had the time to commit to this verification exercise.

REVIEWER COMMENTS

Reviewer #1 (Remarks to the Author):

This ms deals with the problem of assessing the role played by school-related containment measures in curbing the COVID-19 epidemic in the Netherlands. This is achieved by implementing and calibrating a model that includes different age classes and different disease stages. Reproduction numbers are estimated via a Next Generation Matrix approach. This approach was also taken e.g. by Gatto et al. (PNAS, 2020, 117 (19) 10484-10491), but the categories were different locations in Italy, not different age classes. Calibration is performed by fitting data of hospitalizations and seroprevalence, a good choice. The ms is very interesting, as it faces an important issue, that of keeping schools closed or at least partially open. It relies on sound data and the authors are clearly competent in the art of epidemiological modelling. However there are some points of the ms that do require clarification or further analysis.

We thank the Reviewer for finding our study interesting and recognizing that it addresses an important topic and relies on sound data. The revised manuscript owes much to your suggestions. We agree that some points of the manuscript were not explained in enough detail. We have used the opportunity to improve our revised manuscript by giving detailed clarifications and conducting additional analyses. We are also thankful for bringing to our attention the studies by Gatto et al (2020) and Bertuzzo et al (2020). We agree that there are methodological similarities with our study and we have added these papers to References in the revised manuscript.

- The first point concerns the very estimation of the effective reproduction number. The only hint at how R_e is estimated is given at line 172, where the authors state “we evaluated the impact of a control measure by computing the effective reproduction number (R_e) using the next generation matrix method [18, 22].” However, both ref. 18 and 22 specify how to compute R_0 , the basic reproduction number, not the effective reproduction number, if by effective we mean that number at times later than 0, when the prevalence of susceptibles is no longer 1 (Disease-free equilibrium) and the value of the epidemiological parameters is no longer that prevailing at the beginning of the epidemic. Control measures can be accounted for according to 18 and 22, but the calculated R_e is nothing but R_0 in which the parameter values are instantaneously changed at the very beginning when the prevalence of susceptibles is 1. This describes the situation in which the control measures are instantaneously and immediately deployed. I suspect, instead, that the authors have calculated a time-varying next generation matrix which includes both the instantaneous prevalence of susceptibles (<1) and the time-varying values of the epidemiological parameters. Then the spectral radius of the NGM has been calculated at any time. If this is indeed the procedure, I can sympathize with it, even if it is based on practical more than theoretical grounds. However, the authors should a) describe their procedure exactly, b) provide some sort of justification to other literature.

The reviewer is right in that apart from Ref [18] and [22], the original manuscript did not contain the details on R_0 and R_e computation. We have now extended Section *Model analyses* (Lines 394-413) and included Section *Computation of the basic and effective reproduction numbers in Additional information* (Lines 571-650) to explain our procedure. The R_e was indeed computed using the NGM method without considering the decrease in the prevalence of susceptible persons, but taking into account the change in time-dependent model parameters. This is a frequently used procedure that was applied for a number of pathogens including SARS-CoV-2 (see e.g. the recent study by Li et al. Science 2020 [37]). The reason for using it in our study is that the seroprevalence of SARS-CoV-2 in the general population in the Netherlands is very low during the time horizon of our analyses. It was measured at 2.8%, 4.6% and 4.9% in three rounds of the national serological survey conducted in April/May,

June/July and September/October 2020 (<https://www.rivm.nl/pienter-corona-studie/resultaten>). In our study, the corrections to $S=1$ would have a small impact on R_e and would not change the overall conclusions of our study which is focused on the relative impact of school- and non-school-based interventions. To demonstrate this, we have now calculated R_e using the seroprevalence estimates from the 3 serosurvey rounds conducted in the Netherlands and compared it to our original estimates. The corrections for the median R_e are 2.7%, 4.8% and 4.6%, in all cases much smaller than the credible intervals of our estimates.

- The model has been calibrated on the data of the first 69 days after February 27. The parameters, part of which are time varying, are therefore the parameters that describe the variations along these 69 days. Therefore, how did the authors obtain estimates of the effective reproduction number after May 6 (fig. 5)? It is not only a matter of procedure (point above) but also a problem of how the various epidemiological parameters have been estimated, given that the data on hospitalizations and serology refer to the initial period only (line 63 and following). Calculating the time-varying NGM requires that the variations of say the contact rates are estimated. Thus, it is not at all clear how R_e is estimated after May 6 and how its distribution is obtained (Fig. S4C-D). At lines 214-215 the authors claim that $R_e = 1.31$ should be achieved. I do not understand why such a value should be "achieved". Where does 1.31 come from? How was it estimated? Similarly, at line 226 the authors state that in November 2020 (not clear which day) an effective reproduction number of 1.00 had been reached. Again, where does this number come from? At line 264 the authors state that reproduction numbers at different time points of the pandemic correlated well with estimates obtained from independent sources (Dutch dashboard). Given that the model was fit to data in the first 69 days, how were these numbers at different times obtained? Purportedly, by using methods and/or data different from those of the Dutch dashboard. But where are the methods and the data described?

We agree that these methods were not clearly formulated. We have now included explanations in Section *Model analyses* (Lines 394-413) and *Addition information Computation of the basic and effective reproduction numbers* (Lines 571-650). What we did is the following. After fitting the model to the data of the first 69 days, we had posterior distributions of the parameters as shown in Figure S3. As hospitalization data during relaxation is not available, we calibrated the model to values of R_e as published on the dashboard of the National Institute for Public Health and the Environment (RIVM). These time-dependent R_e values are estimated from hospitalization data and later from case numbers using methods described in <https://royalsocietypublishing.org/doi/10.1098/rspb.2006.3754>, Ref [46]. The calibration of the model in the relaxation period is possible because the epidemiological parameters are assumed to be constant throughout the time horizon of the analyses, and only the contact structure as described by g , ω and ζ_2 (constant during the relaxation period) varies with time (see Eq. 3). Since in the relaxation period, schools were open without substantial control measures, ω was fixed at 1 (the proportion of retained school contacts during the relaxation period as compared to the pre-lockdown period). Since there was some decrease in adherence to contact-reduction measures in August and November as compared to April, ζ_2 was fixed at 0.67 (ζ_1 is estimated at 0.51 for April, Figure S3; sensitivity analyses were done for ζ_2). We finally calibrated g such that the median reproduction numbers in the model would equal the specific values estimated by the RIVM (about 1.3 in the period 27 August-6 September and about 1 in the period 7-13 November). Parameter g can be interpreted using Eq. 3. For example, $g = 0.5$ corresponds to half-way in the relaxation of non-school contacts. The distributions shown in Figures S4 C and D are obtained using the NGM method with $\omega = 1$, $\zeta_2 = 0.67$, $g = 0.5$ and $g = 0.8$, respectively, and other parameters drawn from the posterior distributions as shown in Figure S3. In analyses, the parameters ω and g were then used as control parameters to reduce the number of school- and non-school-related contacts (Figures 5, 6 and 7). In doing so, we varied one type of contact and kept the other type constant.

- As a whole, 10 parameters (if I am not wrong, see Table 1) have been estimated. Actually, two of these parameters depend on age (hospitalization rate and susceptibility). As far as I understand the estimated hospitalization rates are 8 (Fig. 3) and the susceptibilities are 3 (ages 0-20, 20-60, 60+). A total of 19 values to be estimated, consequently. Since this number is quite large, it would be interesting to know whether there are correlations (negative or positive) between parameters. If it were so, this would suggest that a more parsimonious model might have been used. For instance, why did the authors choose exactly those 3 categories for susceptibility? I suggest that the estimated correlation matrix between the 19 parameters is reported in the Supplementary Information and commented in the main text. As the authors have implemented a Bayesian Monte Carlo method, it is no problem for them to obtain such a matrix. Some comments on the choice of the number of parameters to be estimated are also necessary.

The reviewer is correct. We estimate 10 parameters (now written down explicitly both in Table 1 and Table 2), two of which are age-dependent (susceptibility and hospitalization rate). In the estimation procedure the susceptibility in the reference age group (60+) is fixed at 1, therefore 18 values are estimated in total. We have now made this clear by adding footnotes in Tables 1 and 2. We also wrote down explicitly the 8 estimated hospitalization rates (Table 2). Regarding the choice of age categories for hospitalization rate and susceptibility (now explained in Section *Observation model and parameter estimation*, Lines 380-393): (i) Due to the low number of hospitalizations in young persons, we assumed that hospitalization rates in the first three age groups (ages [0,20)) were equal, therefore only 8 hospitalization rates were estimated. (ii) We kept the same age categories for the relative susceptibility as in the retrospective cohort study by Jing et al 2020, from where we took the priors, i.e. the relative susceptibility was estimated for ages [0,20) and [20,60), and 60+ age category was used as the reference corresponding to susceptibility equal to 1. We also think it is a natural choice if we want to stratify the population in younger, middle-aged, and older persons.

On the Reviewer's request, we have included the correlation matrix in *Additional information* (Figure S5) and commented on it in the main text. In Section *Observation model and parameter estimation* (Lines 383-384) we wrote "As the age-specific hospitalization rates are positively correlated (Figure S5), we parameterized the model as $v_k = \hat{v}_k \bar{v}$, where \hat{v}_k is a simplex and \bar{v} is a scalar." In Section *Results Epidemic dynamics* (Lines 82-88) we wrote "The joint posterior density of the estimated parameters reveals strong positive and negative correlations between some of the parameters (Figure S5). For instance, the initial fraction of infected individuals (θ) is negatively correlated with the probability of transmission per contact (ϵ) and the hospitalization rate (v_k), as a small initial density can be compensated by a faster growth rate or a larger hospitalization rate. For that reason, the age-specific hospitalization rates are all positively correlated. These correlations highlight the necessity of complementing the hospitalization time series data with seroprevalence data, even if the sample size of the latter is small. Without the seroprevalence data many parameters would be difficult to identify.

Other comments

- The main text of the paper should be made shorter avoiding unnecessary repetitions and delegating part of the technicalities to the supplementary information.

We have restructured the manuscript so that it conforms to the guidelines of Nature Communications. Section *Methods* has been moved to after Section *Results*, and part of the technicalities has been delegated to *Additional information*. The overview of the main points of methods and analyses has been given in the last paragraph of Section *Introduction*.

- The authors have chosen to disregard the spatial variations of the disease, differently from e.g. Gatto et al. (2020) and Bertuzzo et al. (Nature Communications, 11, 4264, 2020) for Italy. This is somehow justified by the high populations density prevailing in the Netherlands

and the large connectivity of their transportation system. A comment on this choice, however, would be welcome.

The main reason for not using a spatial model is that we had access to aggregated country-level data (hospital admissions and seroprevalence). We believe including regional stratification would not result in a better model. Moreover, including stratification both by age and region would result in a model that is computationally less tractable. Following the reviewer's suggestion, we have included a comment on our choice of the model in the revised manuscript and added the suggested references as examples of models which account for spatial variations of covid-19 (Section *Transmission model*, Lines 265-268, newly added references [16,17,37]).

- line 47: “contract” à contact

Corrected.

- line 86: “Inclusion of m identical infectious period etc.” Yes and no. What is Erlang-p distributed is the time between an exposed becoming infectious and a susceptible becoming infected by an individual in the class $I_{k,p}$. So, if one wants an Erlang-m distribution of the infectious period (with $m = 3$ in this ms), one should implement a model in which a susceptible is infected only by individuals in the class $I_{k,m}$. The other classes are purely fictitious and serve only the purpose of generating the desired Erlang-m distribution. If all the classes $I_{k,p}$ are assumed to be equally infectious (parameter β_k) then the infectious period is distributed as the sum of Erlang-1 plus Erlang-2 etc. I am afraid that the paper D. Champredon, J. Dushoff, D.J.D. Earn, Equivalence of the Erlang-distributed SEIR epidemic model and the renewal equation, SIAM J. Appl. Math. 78(2018) is frequently misunderstood. What Champredon et al. call the Erlang-distributed SEIR epidemic model does not have an Erlang distributed generation interval (as the authors actually show in their sect.3)! All this is related to hyper-Erlang distributions in queueing systems. I think that the present ms should be amended in this respect.

We are very familiar with the paper by Champredon et al which is now cited together with a very instructive paper by Diekmann et al (Line 279 [38,39]). There seems to be a misunderstanding as we are not claiming that the generation time is Erlang-distributed but only that the infectious period (the time spent in the I compartment) is (Line 278). Our model is similar to Eq 2.1 in Champredon et al who mention in their introduction: "A generalization of this model, which we refer to as the Erlang SEIR model, divides the E and I stages into m and n substages, respectively. All m latent (resp., n infectious) substages are identical. This subdivision is usually viewed as a mathematical trick in order to make latent and infectious period distributions more realistic; the resulting latent and infectious periods have Erlang distributions (gamma distributions with integer shape parameter)". There also seems to be confusion regarding infectiousness of persons in different infectious compartments. We assume infectiousness is the same for all $I_{k,p}$ compartments (Line 274). We have now added these explanations in the model description (Section *Transmission model*, Lines 264-288).

- Legend of Fig. 1: Authors should specify that the number of contacts is actually “number of contacts per day”.

The reviewer is correct. We have changed “number of contacts” to “number of contacts per day” in the caption and in the legends of Fig. 8 B, C and D (old Fig. 1).

- Line 100: Here the authors might specify that contact rate is actually measured as number of close contacts per day. I assume that the authors by contacts mean “close” contacts. Maybe they should better explain this point.

The text has been updated as requested (Section *Transmission model*, Lines 295-300).

“We denote the general contact rate (the number of contacts per day) of a person [...]. The contacts are defined as contacts with household members and contacts in the community [19]. Examples of a contact outside one's household are talking to someone (face-to-face), touching someone, kissing someone or doing sports with someone. More details on contact matrices before and after the lockdown can be found in [19].”

- Line 114 (eq. 3): I think that the choice of subtracting school contacts from pre-lockdown contacts is not very clear. The authors should clarify this point.

Thank you for pointing out that the rationale behind Eq. 3 was not clear. We have now adjusted the text to explain this equation in more detail (Section *Transmission model*, Lines 310-324). To investigate the impact of school- and non-school-based measures individually, we need to be able to split the contact rate into a rate of contacts occurring at schools and a rate of contacts occurring elsewhere. The contact rates we used from the literature are additive [19,20], thus the contact rate before the lockdown (b_{kl}) can be written as a sum of the school contact rate at the pre-lockdown level (s_{kl} , see Fig. 8 D) and the contact rate for all locations but schools ($b_{kl} - s_{kl}$). The contact rate after the lockdown (a_{kl}) by definition did not include any school contacts because all schools were closed. In Eq. 3 we formulate how the contact rate changes after the relaxation of control measures after the first lockdown due to (i) increase of non-school contacts from the level after the first lockdown (a_{kl}) to their pre-lockdown level ($b_{kl} - s_{kl}$) and (ii) opening of schools which we assume to happen instantaneously (s_{kl}).

- Line 134: “ $N_k = S_k$ etc.” We cannot think that hospitalized (and quarantined, not modelled here) individuals are mobile. The correct force of infection is that reported in Gatto_et_al (2020) and Bertuzzo_et_al (2020). It probably does not make such a big difference in practice, but conceptually the calculation of N_k is wrong. Also, the assumption of frequency-dependent force of infection, rather than e.g. density-dependent, should be commented.

Thank you for this comment. We have added explanation in Section *Model equations* (Lines 337-350, Lines 283-285) to address your concerns. Unlike in e.g. Gatto et al (2020), in our model, the denominator in the force of infection (N_k) indeed includes hospitalized persons, $H_k(t)$. Our variable $H_k(t)$, however, describes the cumulative number of hospital admissions at time t (as opposed to the current number of hospitalized persons at time t in Gatto et al (2020)). If hospitalized persons would not be involved in any contact, we agree that it would make sense to subtract the current number of hospitalized persons (not the cumulative) from N_k . As the reviewer points out, this will have a very small effect as the number of hospital beds is much smaller than the population size. On the other hand, hospitalized persons will be involved in some contacts with medical personnel and visitors and, hence, the ‘true’ value of the denominator will be value between N_k and the value suggested by the reviewer. As the choice will not impact the results, we decided to stick to our model, primarily to avoid the introduction of new notation of the ‘current number of hospitalized persons’. Because we assume that contacts of the currently hospitalized persons (who may still be infectious) will not be infected due to the use of personal protective measures by medical personnel and hospital visitors, the ‘current number of hospitalized persons’ does not contribute to the force of infection. As patients who are discharged and recovered (or deceased) also do not contribute to the force of infection, the ‘cumulative number of hospitalized persons’ ($H_k(t)$) does not contribute to the force of infection either.

Like many other authors, we indeed assumed a frequency-dependent transmission where the per capita rate at which a susceptible individual becomes infected increases with the fraction of the population that is infectious. We think this choice is justified for the Netherlands as one of the most densely populated countries in Europe. Moreover, as the population size will not change substantially during the simulated time period, there is no difference in the outcome

between a frequency-dependent and a density dependent model once the parameters are fitted to obtain the observed reproduction number.

- Line 157: “Parameters were estimated in a Bayesian framework using methods we developed before”. Please give the reader at least a hint about the nature of these methods.

The details of the method were given at the end of the paragraph starting at Line 157 in the original manuscript. To make our text clearer, we changed the order of the sentences in the paragraph (Lines 376-379).

- Legend of Fig.2: Specify the starting date (27 February day 0 ?).

We have added to the Legend of Fig. 1 (old Fig. 2) the following sentence

“Day 1 corresponds to 22 February 2020 which is 5 days prior to the first officially notified case in the Netherlands (27 February 2020).”

- Line 322: “sunsequent” à “subsequent”

Corrected.

Reviewer #2 (Remarks to the Author):

This is a standard age-stratified SEIR model to assess the impact of school closure in the Netherlands. The model was parameterized based on backward inference from empirical hospitalization and seroprevalence data.

We thank the reviewer for the valuable comments and giving us the opportunity to revise our manuscript. We have addressed the comments one by one below.

I am concerned about the treatment of the two key empirical datasets without due regard to potential biases in the way they were assembled. Time varying (depending on pandemic stage) factors that would have affected case ascertainment, identification through testing and admission threshold, in addition to laboratory details regarding the veracity/validity of serosurveys should have been explicated and taken into fuller account in the modelling.

We would like to point out that hospital admission data we used is the most untouched by time trends (as opposed to case notification data where significant changes have occurred during the course of the pandemic). With regard to time varying factors that would have affected case ascertainment, we have added the following (Section *Data*, Lines 242-247):

“The criteria for hospital admission have not changed during the pandemic, and from the early stages all hospitalized patients with a clinical suspicion of COVID-19 were tested by RT-PCR. In all stages of the pandemic, patients requiring hospital admission were hospitalized and the practice of not referring patients for hospital admission (e.g. due to self-expressed treatment restrictions or moribund condition) did not change.”

Regarding the seroprevalence data we have added the following (Section *Data*, Lines 248-256):

“Participants for the serosurvey were enrolled from a previously established nationwide serosurveillance study, provided a self-collected fingerstick blood sample and completed a questionnaire. IgG antibodies targeted against the spike S1-protein of SARS-CoV-2 were

quantified using a validated multipleximmunoassay. Seroprevalence was estimated controlling for survey design, individual pre-pandemic concentration, and test performance.”

More details on the seroprevalence data used in our work is given in [18]. We further explained how potential biases in this dataset could affect our conclusions as follows (Section *Discussion*, Lines 165-170):

“However, even with extensive validation, we need to be careful when interpreting the predictions of our model as these depend on the sensitivity of serology to identify individuals with prior infection. Recent studies suggest that in persons who experience mild or asymptomatic infections, SARS-CoV-2 antibodies may not always be detectable post-infection [27,28]. Therefore, more children may have had an infection than indicated by the seroprevalence survey because the proportion of asymptomatic in children is believed to be high. As a consequence, our study potentially underestimates the role of children in transmission.”

The manuscript is written in such a way that it does not highlight the salient points of the background, methods or findings. It reads more like a detailed log.

We have restructured the manuscript so that it conforms to the guidelines of Nature Communications. Section *Methods* has been moved to after Section *Results*, and part of the technicalities has been delegated to *Additional information*. The overview of the main points of methods and analyses has been given in the last paragraph of Section *Introduction*.

The authors should explain how they treated importation seeding, and why or why not.

Importation of new cases was very important at the beginning of the pandemic when the population was fully susceptible and there were no control measures in place. To account for this, we estimated the fraction of infected individuals (in exposed and infectious states) 5 days prior to the first official case in the Netherlands (t_0) by fitting the model to the data. In later stages of the pandemic, importations do not play such an important role because of existing pool of infectious individuals within the country and ongoing control measures. For this reason, importations after t_0 were not included in the model. We have now made this clear in the last paragraph of Section *Model equations of Methods* (Lines 561-358).

Model details should be checked and the code vetted – I have not had the time to commit to this verification exercise.

All scripts reproducing the results of this study are available in the GitHub repository, <https://github.com/lynxgav/COVID19-schools> (see Section *Code availability*, Lines 421-423), and can be checked at the Reviewer’s convenience. All datasets, except for hospital data, analysed and generated during this study are available in the same GitHub repository (see Section *Data availability*, Lines 416-420). The applications for the hospital dataset should be forwarded to the National Institute for Public Health and the Environment but we have shared this dataset with the Editor so that the reviewer can complete necessary checks.

Reviewers' Comments:

Reviewer #1:

Remarks to the Author:

The authors have modified the manuscript according to my requests. It is now much clearer and better organized. In my opinion, this revised version can be published.

Reviewer #1 (Remarks to the Author):

The authors have modified the manuscript according to my requests. It is now much clearer and better organized. In my opinion, this revised version can be published.

The authors wish to thank this Reviewer for their valuable comments and favourable assessment.